# Cost Valuation and Climate Mitigation Impacts of Forest Management: A Case Study from Piatra Craiului National Park, Romania

Serban Chivulescu [1], Raul Gheorghe Radu [1,*], Florin Capalb [1,2], Mihai Hapa [1,2], Diana Pitar [1], Luminita Marmureanu [1], Stefan Leca [1], Stefan Petrea [1,2] and Ovidiu Badea [1,2]

[1]   National Institute for Research and Development in Forestry "Marin Drăcea", 128 Eroilor Boulevard, 077190 Voluntari, Romania; serban.chivulescu@gmail.com (S.C.); florin.capalb@yahoo.com (F.C.); ionutmihaihapa@gmail.com (M.H.); diana.silaghi@icas.ro (D.P.); luminita.marmureanu@gmail.com (L.M.); fane.leca@gmail.com (S.L.); stefan.petrea19@gmail.com (S.P.); ovidiu.badea63@gmail.com (O.B.)

[2]   Faculty of Silviculture and Forest Engineering, "Transilvania" University of Brașov, Șirul Beethoven 1, 500123 Brașov, Romania

*   Correspondence: raulgradu@gmail.com

**Abstract:** With the intensification of the effects of climate change, the urgent need to address their drivers, especially greenhouse gas emissions, has become essential. In this context, forests offer a robust solution, with their potential to store and mitigate carbon emissions. However, striking a balance is critical given the significant economic contribution of the forestry and wood-based industries, which account for about 5% of Romania's GDP and employ 6% (around 300 thousand) of its active workforce. This study, conducted in the Piatra Craiului National Park located in Romania's Southern Carpathians, we utilize the EFISCEN application to generate three distinct 50-year forest evolution scenarios based on harvest intensity, namely Business As Usual (BAU), Maximum Intensity (MAX), and No Harvest (MIN), on two historical different managed forests, i.e., conservation and production. The study aims to guide forest owners in decision making with scenario modeling tools, with the objectives of assessing the forest carbon sequestration potential and evaluating the economic feasibility. In the most probable scenario, the BAU scenario, the growing stock increases from 2.6 million $m^3$ to 3.8 million $m^3$ over 50 years, with a more than 40% increase. Comparing the carbon stock change for all tree harvest scenario types indicates that the MIN scenario has the highest carbon sink capacity in the next 50 years; the BAU scenario is a well-balanced option between carbon sink and wood provision and has an optimal EUR 3.7 million in annual revenue. The MAX scenario can boost the growth and increase the annual revenue from wood by 35% but is effective only for a short time and thus has the smallest calculated revenue in time. Achieving a win–win relationship between carbon sequestration and wood supply is imperative, as well as good planning and scenarios to contribute to climate mitigation and also as provisions for local communities and to sustain the local economy.

**Keywords:** climate change effects; peri-urban forest management; forest stability; urban expansion; greenhouse gas emissions; forest resilience





## 1. Introduction

Climate change and air pollution have emerged as one of the most pressing global challenges affecting human health, ecosystems, and biodiversity [1,2]. Awareness has intensified and rapidly increased worldwide [3,4], with the intensification of extreme weather events and rising global temperatures signaling the urgent need for effective mitigation strategies, primarily through the reduction of greenhouse gases [5,6]. In this context, forest ecosystems play a key role, serving as both sinks and greenhouse gas sources [7]. The dual ability of forests to store carbon and release it back into the atmosphere situates them

at the heart of climate change mitigation efforts [8–12]. An important component in the sustainable type of forest management [13–15] is represented by the carbon cycle [16–18], which contributes to the ongoing changes to the state of forest ecosystems [19–22]. The intensification of climate change [23] and the carbon cycle have led to conflicting interests and incomplete knowledge among stakeholders [24]; thus, the efforts of mitigation have been increased [25–28], mainly in the form of collective actions [29,30]. While the global importance of forest management in climate change mitigation is well recognized [31], the specific impact of different management strategies on carbon sequestration and economic outcomes remains an area needing further exploration. This is particularly true for regions like the Piatra Craiului National Park (PCNP) in Romania, where the balance between conservation efforts and the economic contributions of the wood industry is delicate and consequential. The wood industry plays a significant role in Romania's economy, contributing in 2021 to 4.6% of the national GDP and employing 6% of its active workforce [32]. Simultaneously, forests like those in PCNP are vital carbon sinks, offering significant potential for climate change mitigation. This comparison presents a complex scenario where forest management decisions have far-reaching implications for both the environment and the economy. Attributing economic value to the beneficial effects that forest carbon storage brings is an important and challenging aspect in managing forest ecosystems [33–38]. Research on the economic evaluation of the carbon stored by forests has identified many challenges [39,40], underlining the fact that the price of carbon taxation is dynamic and depends on the impact that they have, at that time, on climate change and humanity's desire to limit the effects produced by it [41]. Currently, greenhouse gas emission certificates, also called carbon certificates or $CO_2$ certificates, are traded on the relevant markets [42]. This study aims to fill the research gap by providing a nuanced understanding of how forest management scenarios impact carbon sequestration and economic outcomes in the PCNP.

The European Forest Information SCENario Model (EFISCEN) is a potent tool for the modeling of carbon sinks and fluxes across varied biomass compartments, such as living biomass, dead wood, litter, and soil [43]. Originally designed as a large-scale model to project forest resources at the European level [44], the EFISCEN has since proven its adaptability for regional analyses in response to localized decision-making needs for future forest management [45]. This versatility is supported by studies conducted in various contexts, including Finland [46] and Romania [47].

By employing the EFISCEN application, we generated three distinct 50-year forest evolution scenarios based on harvest intensity: Business As Usual (BAU), Maximum Intensity (MAX), and No Harvest (MIN). These scenarios were applied to two historically different managed forests, i.e., conservation and production, to gauge their respective impacts.

The growing necessity of mitigation strategies in forest management highlights the need for innovative methods that consider the economic impact of these decisions. This study aims to provide a step-by-step example for forest owners to make informed choices using scenario modeling tools. It evaluates the economic consequences to help balance decisions between conservation and the silvicultural management of forest areas.

Our study seeks to answer two primary questions: (1) What is the carbon removal potential of forests in the national park for the next decade under different management scenarios? (2) Can the most suitable scenario for carbon removal be implemented in the future, considering the economic implications?

By addressing these questions, this research aims to contribute valuable insights to the broader climate change mitigation and forest management field. It seeks to inform policy decisions and management practices so as to achieve a sustainable balance between environmental conservation and economic viability.

## 2. Materials and Methods

### 2.1. Location and Description of the Studied Site

The research was conducted in the Piatra Craiului National Park, located in the Southern Carpathian Mountains in the central area of Romania (Figure 1). The park boundaries cover an area of 14,700 ha, while the area covered with forest vegetation is approximately 11,400 ha [48]. The forest land use covering the park area is included in a management study where the forest is stratified according to the forest management practices, topography, climate, and ecosystem type. Forest vegetation is defined as an area covered by forest vegetation larger than 0.25 hectares, with a minimum tree height of 5 m at maturity and canopy cover over 10 percent and wider than 20 m, which can include areas that temporarily do not meet the above minimum thresholds but are expected to reach them in the future [48].

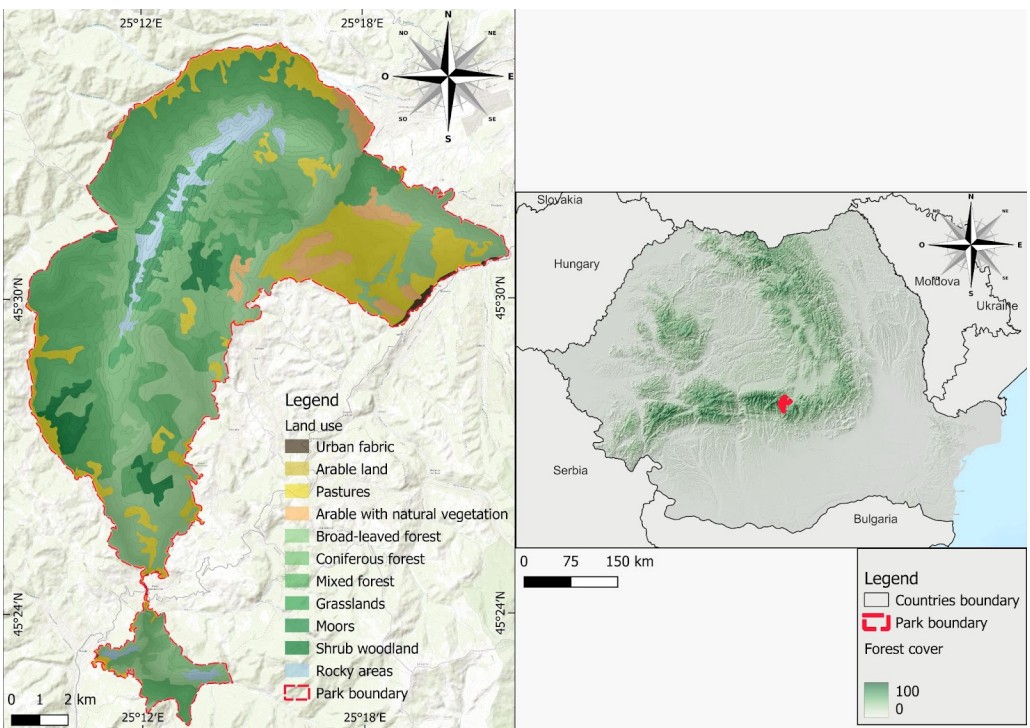

**Figure 1.** Location of the study site in the center area of Romania (**right**) and PCNP boundaries (**left**).

The area's climate is characterized by a temperate continental type in which two topo climates have been identified. The first characteristic is the high mountains, and the second is the low mountains. The average annual amount of precipitation varies between 800 and 1200 mm, and the average annual temperature has values between 0 and 5 °C, values assessed for 1961–2000. Snow falls between November and April, with a maximum in January and February, when the heaviest snow is recorded [49].

The forest vegetation in the research area consists mainly of spruce, beech, and beech–spruce mixtures. These forests were established as a national park for biodiversity conservation purposes in 1990 [48]. At the same time, these biodiversity conservation interests include the socio-economic development interests of the local communities. The research area includes a strict protection area (43%), an integral protection area (1%), a sustainable conservation area (46%), and a sustainable development area (10%). Forest areas included in the park support various management regimes, strictly regulated by the technical assessment made in a management plan every ten years. Nevertheless, all the management decisions align with the area's conservation management principles. Only special conservation treatments are allowed in the protection area, involving a minimal wood extraction volume (less than 12% of the total volume in a decade). This is permitted only when

stands are affected by various factors such as windthrow, insect attacks, or diseases. In the sustainable conservation area, continuous cover and selection systems are permitted; mostly shelterwood and sanitary felling is performed in the sustainable development area. Although the park's area overlaps with several Natura 2000 sites, its management is not influenced by the presence of this network.

### 2.2. Overview of Carbon Sink and Wood Supply Model

The EFISCEN model is an easy to use and comprehensive model primarily developed for the estimation of carbon stock changes over extensive regions. Nevertheless, its scalability allows for the effective prediction of biomass stock across various forest compartments, even with limited initial data, such as the age class distribution, growing stock, and growth rates in a specific region. At the same time, it can also accommodate more complex applications, including diverse climate scenarios, harvesting intensities, and species mixtures. It requires forest structure data, such as the area, average stand growing stock volume, and annual increment, which are further stratified by age class and species. Essential supplementary data, including biomass distribution parameters, the turnover rates of biomass components, litter quality, climate parameters, thinning details, and final felling information, are also requisites for the model. With the capacity to estimate parameters like the growing stock, increment, standing dead wood, harvest level, and age class distribution over time, the granularity of the input data modulates the model's predictions. Thus, it can offer an array of forest resource estimates based on diverse scenarios defined by management types.

A unique feature of the EFISCEN model is its ability to estimate the carbon stock content within the dissolved organic matter (DOM) and the carbon content within the soil organic carbon (SOC) pool in forest soils. This is facilitated by the YASSO module integrated into the primary software, which considers input from trees and manages the distribution between tree biomass and litter and soil decomposition rates. For users, the EFISCEN manual offers both default and recommended values for these parameters [43].

In our study, we used the EFISCEN model [50], 4.2.0-19 version [43], to simulate the evolution of forest resources within the PCNP.

### 2.3. Model Initialization and Simulation

Forest districts in Romania boast a rich history of maintaining ecological, economic, and social functions. These forest districts undergo an inventory every 10 years, aligning with a dedicated management plan. The first forest management plan for the PNPC was created in 1952 [48], and, today, the forested area within the national park is divided into approximately 1426 individual forest stands. Each stand represents a uniform type of forest stratum characterized by its species, age class, growing stock, increment, topography (slope exposure), microclimate type, and forest regime [51]. The latter pertains to regulating the stand's production process concerning its regeneration method, whether through seed, slash, or a mixed approach.

To facilitate this classification, we sourced data from the management plan database for each forest stand. These data encompassed the area, average age, species composition, average stand growing stock volume, annual net increment, harvest demand, felling and thinning age, and regeneration period. The management plan provided all this information through field inventories, which are mandatory every ten years. The term "stand volume" here refers to the entire above-ground volume of trees larger than 8 cm in diameter at breast height (dbh), including the stem, bark, and branches exceeding 5 cm in diameter. Tree volume calculations are derived from field observations and measurements—specifically, the tree species, diameter at breast height (DBH) (in centimeters), and tree height (H) (in meters). Direct observations determined the species of the trees, the DBH was measured with a tree caliper, and the height was measured with the help of a hypsometer. These measurements were then applied to the bifactorial volume regression Equation (1). The

regression coefficients for this equation have been determined for 43 forest species, as described by Giurgiu [52].

$$log\ v = a_0 + a_1\ log\ d + a_2\ log_2\ d + a_3\ log\ h + a_4\ log_2\ h \tag{1}$$

where:

$d$ represents the diameter of breast height (cm);

$h$—tree height (m);

$v$—tree volume ($m^3$).

We initially grouped the collected data into management-type units and group species to initialize the model. We provided the age class distribution of area, volume, and growth to generate the input tables (Supplementary S2 (1) Model initialization data). This grouping was based on historical harvest statistics and the prevailing forest regime. The forest area in the PCNP, totaling 11,400 ha, can be stratified based on the management types: 7700 ha for forest conservation stands and 3700 ha for wood production forest stands. Each forest stratum was further categorized into three species groups: coniferous (CO), beech (BE), and mixed stands. These categorizations were formed by amalgamating similar forest stands for which growth curves were processed (Figure 2A,B).

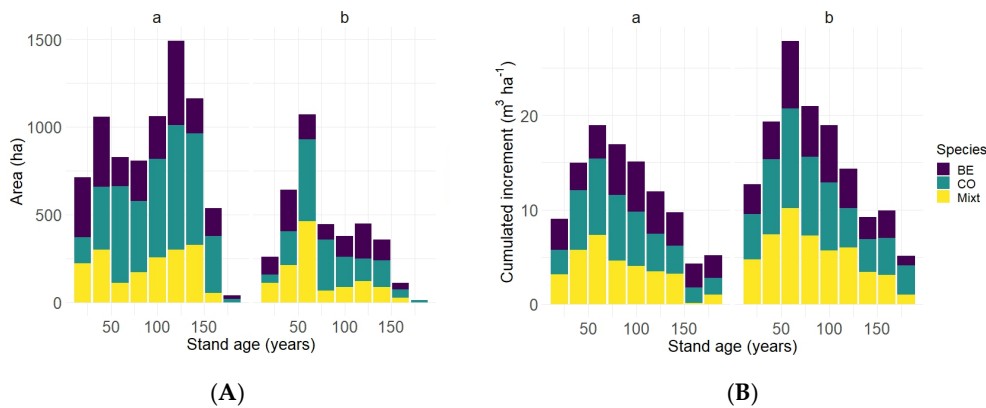

**Figure 2.** (**A**) Forest area according to historical management (conservation stands—left [a], production stands—right [b]) and group species for nine age classes. (**B**) The average net annual increment according to group species and age classes for the two historic management scenarios (conservation stands—left [a], production stands—right [b]).

By preprocessing the raw data from the management plans, we obtained the information necessary as input for the EFISCEN model. This model uses data for each forest stratum, such as the area, average growing stock, increment for each age class, and harvest intensities. Outputs were generated for six forest types in the PCNP, spanning two management types (conservation and production forests) and three species groups (BE, CO, and Mixed). The initial matrices required by the model for initialization were crafted using the P-2009 aditional software tool part of the EFISCEN 4.20 [43].

**Carbon stocks in living biomass.** The estimation of carbon stocks within living biomass is based on the initial volumes of the growing stock for each species group, as per the IPCC guidelines [53]. Data regarding the proportion of biomass components, wood densities, and root ratios were sourced from national studies [52], while the carbon fraction was referenced from the IPCC [53]. The equation below was used to estimate the annual increase in carbon content for both above-ground and below-ground biomass:

$$\Delta C = A \times I_V \times D \times BEF \times (1 + R) \times CF \tag{2}$$

$$\Delta CO_2 = \Delta C \times (-MW\ of\ CO_2/MW\ of\ C) \tag{3}$$

where:

$\Delta$C—annual increase in above-ground and below-ground biomass carbon stock due to biomass growth, tonnes C $yr^{-1}$;

A—forest area in each stratum, ha;

$I_V$—net above-ground volume increment, excluding mortality volume estimated over bark, including all branches with at least 5 cm diameter, $m^3ha^{-1}yr^{-1}$;

D—basic wood density, tonnes d.m. $m^{-3}$, derived from national studies;

R—ratio of below-ground biomass to above-ground biomass, dimensionless;

BEF—biomass expansion factor;

CF—carbon fraction of dry matter, tonne C;

MW of $CO_2$/MW of C—ratio of molecular weight of $CO_2$ (44) to that of carbon (12), approximately 3.67.

$I_V$, the average annual volume increment, retrieved from the stand, was used to fit the growth dynamics function for a five-year step according to the function below [54]:

$$I_{vf}(T) = a_0 + a_1/T + a_2/T^2 \tag{4}$$

where:

$I_{vf}$—the five-year volume increment as a percentage of the growing stock;

T—age of stand in years;

$a_0$, $a_1$, $a_2$—coefficients (dimensionless). The coefficients were fitted using the R software version 4.3.2 [55] based on stand data.

Mortality rates were computed from the local harvest statistics, which referred to sanitary cutting. The volumes estimated for broadleaf species, as found in the forest management plan database, corresponded to the total volume of trees above the ground. As a result, a factor of 1 was applied. However, for conifers, the biomass expansion factor (BEF) was employed to deduce the complete above-ground volume from the merchantable volume available, drawing values from Table 3A.1.0 of the Good Practice Guidance [56]. Regarding wood density (D), there are country-specific values for 19 tree species, as cited by Giurgiu [52]. Using these values, average densities for different species groups were calculated for the modeling task (Table 1). The carbon fraction (CF) of dry matter followed the default values set by the IPCC 2006 GL [56], which are 0.48 for conifers and 0.51 for broadleaves, measured in tonnes C per tonne dry matter $(t.d.m)^{-1}$. The root-to-shoot factor (R) was used variably across specific species groups, applying country-specific factors as mentioned in [52].

**Table 1.** Average values for wood density, root-to-shoot factor, biomass expansion factor, and carbon fraction. These values were instrumental in estimating the carbon stock derived from wood volumes.

| Species Groups | Wood Density Tonnes d.m.$^{-1}$ | Root-to-Shoot Factor Value (R) Dimensionless | Biomass Expansion Factor (BEF) | Carbon Fraction (CF) |
|---|---|---|---|---|
| Conifers | 0.400 | 0.19 | 1.15 | 0.48 |
| Beech | 0.550 | 0.21 | 1 | 0.51 |
| Mixed stands | 0.500 | 0.20 | 1.075 | 0.50 |

**Carbon stocks in DOM and SOC.** We utilized the YASSO [56] decomposition sub-model incorporated into the EFISCEN modeling software version 4.20 to estimate the stocks of dead organic matter and soil organic carbon. This sub-model projects soil carbon stocks, inclusive of litter compartments. Litter production is derived from the yearly biomass, considering the proportions of stems, branches, roots, and leaves as turnover rates, and includes harvest residues transitioned into litter. The soil module details carbon content across three litter and five decomposition compartments [57], influenced by the climatic conditions. We adhered to the suggested parameters [58] for the soil carbon module,

setting the local average annual temperature at 3 °C and accounting for a mean disparity of −50 mm between precipitation and potential evaporation during the vegetation phase (May–September) for all management divisions.

**Management scenario.** The EFISCEN model can simulate a user-defined regime by adjusting the harvest intensities or aligning with a specific region's harvest demand. We employed the model to examine carbon stock trajectories under three harvest intensity scenarios: Business As Usual (BAU), Maximum Intensity (MAX), and No Harvest (MIN).

The forest management planning activity is conducted according to national legislation and current regulations, and its implementation is closely monitored, particularly by the district forest authority and also by the Ministry of the Environment, Water and Forests of Romania. We can specify that the activities of forest management plans are equally guided by the Nature Protection Law (137/2008) and the Forestry Law (46/2008), with subsequent additions, because the two laws are consistent with each other.

Importantly, the BAU scenario in our study accurately reflects the historical management practices employed in the **PCNP**. This scenario provides valuable insights into the park's past management approaches. Per the management plans for conservation-focused forest units, the permissible cuts prioritize sanitary and conservation silvicultural felling, promoting forest ecosystem health with as minimal intensity as practical. In units designated for wood production, the prescribed thinning age spans 40–80 years for all species. The final cut is set at 110 years for conifers and mixed stands, and 120 years for beech. The BAU scenario was framed using historical management practices to outline the bounds for felling and thinning, projected over the next 50 years based on the existing growing stock. Information on the felling age, regeneration time frame, and thinning intensity was sourced from the management plans, which also charted practices for the upcoming decade. This ensured that the management regime, as characterized in the model, mirrored the definitions within the management plans as closely as possible. The MAX scenario assumes that all the wood suitable for thinning and final harvesting operations is utilized, modeling felling intensities to align with the annual growth increment. The thinning period and final harvest cycle are defined based on the historical silvicultural operations data in the management plan. It was designed to match the maximum achievable harvest. The MIN scenario assumes no wood harvest.

### 2.4. Valuation of Carbon Credits and Wood Supply

The EFISCEN model was employed to estimate the wood supply and carbon stocks, assessing these parameters for different types of forest management while considering the average cost–benefit ratio. For the forests in the Piatra Craiului Park, carbon revenue was generated through carbon credits, reflecting the reduced greenhouse gas emissions resulting from management activities. These credits were traded within the European Union Emissions Trading System (EU ETS), a cornerstone of the EU's climate strategy aimed at curbing greenhouse gas emissions cost-effectively. Carbon credit prices were obtained from the European Energy Exchange (EEX), a key influential trading platform for energy and commodity products, including the carbon allowance cap of the EU ETS (Supplementary S1). Both indicators, namely the price of carbon credits and the forecasted quantities of European wood and carbon, show an upward trend. For the purpose of this analysis, 2022 prices were used. The average price applied for a tonne of carbon was EUR 80, and, for wood, EUR 60 per cubic meter was considered. These figures are derived from the referenced Thomson Reuters report on carbon pricing [59] and the Romanian National Statistics Institute [32].

### 3. Results

The simulation model is built on historical information, regarding the age class structure of the growing stock and thinning and harvest intensities to achieve a realistic simulation. The forecasting model runs for a reasonable simulation period of 50 years (2022–2072) for all forest stands in the PCNP. This scenario simulates the aging, growth, and loss of

forest biomass due to mortality and harvesting, and the analyses were carried out for both types of management studies and forest conservation and timber production, respectively.

For the BAU scenario, the model simulation shows, for both management types, that the majority of the forest area and stock will belong to the age class above 100 years, with particularity for the forest conservation scenario, which shows that more than 2000 ha (about 15% of the park area) is covered with forest older than 150 years (Figure 3A,B), thereby contributing to the overall ecological stability of these ecosystems. The increase in very old stands, within the 160 years and older age class, is attributable to the absence of silvicultural treatments. Given the dual management approach within the research area, it was imperative to consider this aspect in the aging simulation.

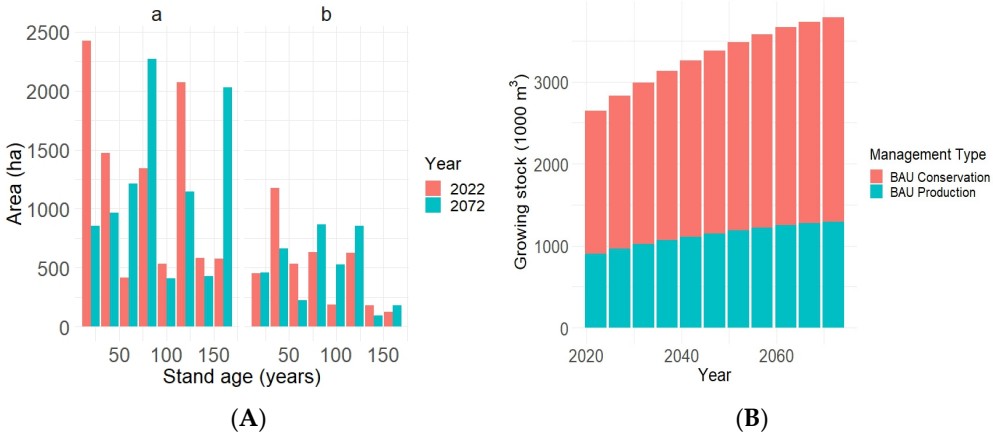

**Figure 3.** Forecasting simulation in PCNP, using EFISCEN model for conservation and production scenarios in the period 2022–2072 as follows: (**A**) forest age distribution for (a) conservation (b) production; (**B**) forest stand growing stock.

The model forecasts a decline in the average annual growth rate. Starting at 5 cubic meters per hectare per year, it is expected to drop by one cubic meter, marking a 20% reduction over 50 years (Figure 4). In contrast, the conservation management scenario predicts a more moderate average growth rate: 3.3 cubic meters per hectare per year, with only a 10% decline over the same period. The growth rates for the next half-century are also influenced by the age dynamics of each tree species. Notably, beech trees tend to grow slower than conifers under both management strategies.

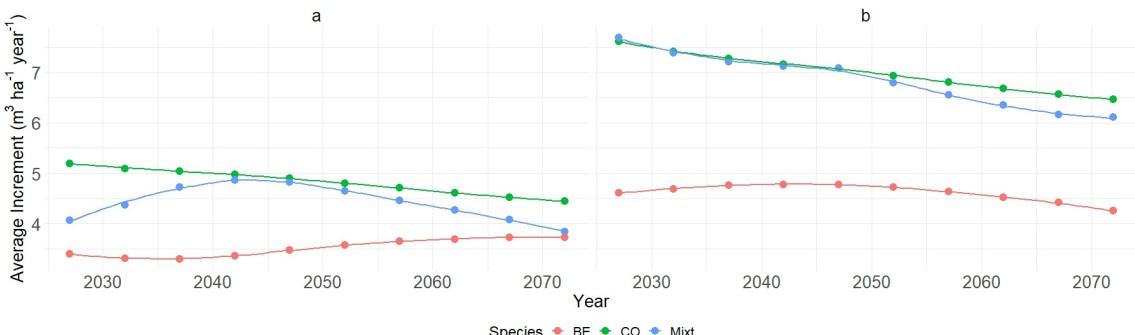

**Figure 4.** Temporal variation in average stand increment (IncrAV) (m³/ha/yr) in PCNP: conservation (left panel (**a**)) and production (right panel (**b**)) scenarios for beech (BE), coniferous (CO), and mixed (Mixt) forests (2022–2072). The line in the figure represents a summary of trends within the data, employing locally estimated scatterplot smoothing (LOESS), a non-parametric method for smoothing scatterplots.

As forest stands age, it is expected that higher mortality rates will occur. This trend is reflected in our model, which predicts a noticeable increase in the volume of standing dead

wood: from 87,000 cubic meters (Figure 5) at the start of the simulation to 124,000 cubic meters by 2072. This increase can be mainly attributed to natural aging and associated mortality, indicating a significant shift in the ecosystem's composition over time. In the production scenario, however, a significant portion of this dead wood will be removed through forestry operations.

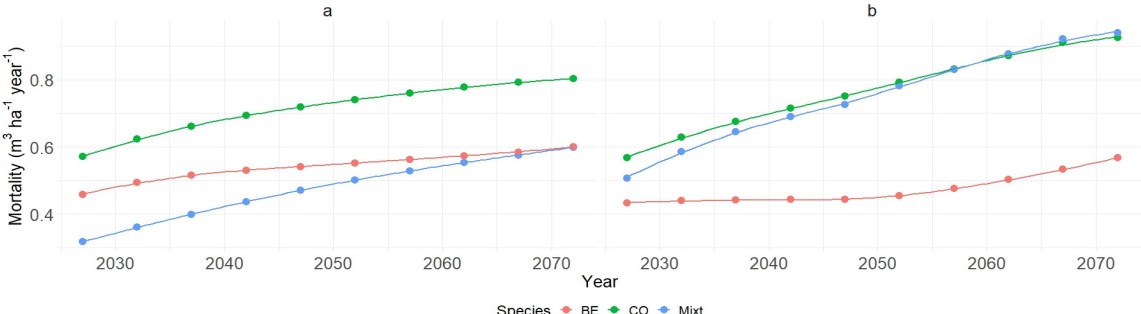

**Figure 5.** Temporal variation in natural mortality volume (Nat mort ha) (m³/ha/yr) in PCNP: conservation (left panel (**a**)) and production (right panel (**b**)) scenarios for beech (BE), coniferous (CO), and mixed (Mixt) forests (2022–2072) using EFISCEN prediction models. The line in the figure represents a summary of trends within the data, employing locally estimated scatterplot smoothing (LOESS), a non-parametric method for smoothing scatterplots.

We also foresee an uptick in timber volume ready for harvest (Figure 6). This is because many forest stands will mature, reaching their optimal age for harvesting and thus ending their growth cycle. The volume of wood from secondary treatments, like thinning, is also expected to grow slightly, representing about 10% of the available stock by 2072. This equates to an average annual thinning volume of around 5700 cubic meters.

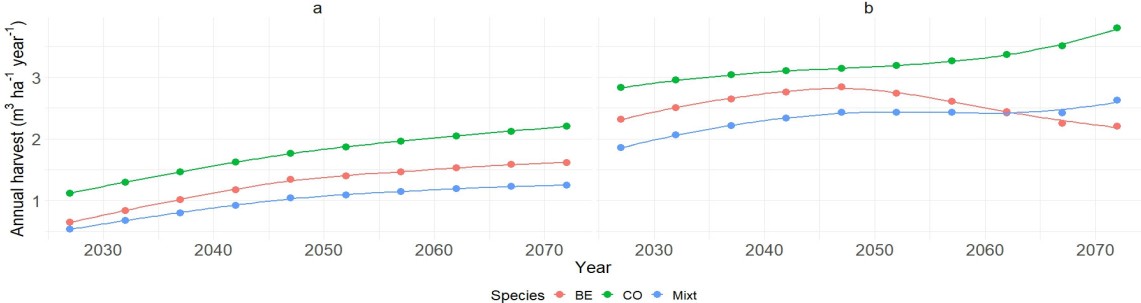

**Figure 6.** Temporal variation in available average wood annual harvest (m³/ha/yr) in PCNP: conservation (left panel (**a**)) and production scenarios (right panel (**b**)) for beech (BE), coniferous (CO), and mixed (Mixt) forests (2022–2072) using EFISCEN models. The line in the figure represents a summary of trends within the data, employing locally estimated scatterplot smoothing (LOESS), a non-parametric method for smoothing scatterplots.

The wood volume set aside for final harvest is estimated at an average of 17,600 cubic meters annually. However, there are fluctuations over the 50-year simulation, leading to an estimated 80% increase in the available volume for harvest by its end. This variability in harvestable wood volume should provide greater financial benefits to forest owners, aligning with forestry policies prioritizing the maintenance of forest continuity and encouraging natural tree regrowth.

The projected total growing stock is anticipated to rise from 2.6 million cubic meters in 2022 to 3.7 million cubic meters by 2072 in the Business As Usual (BAU) scenario (Figure 7a). This increase in growing stock is a balance of the net result of tree growth, mortality, and annual harvesting. The average annual net increment is 4.6 cubic meters per hectare, with average annual mortality of 0.6 cubic meters per hectare. The projected annual average

volume extracted from the forest is around 2 cubic meters per hectare per year, leading to a net average increase of 2.6 cubic meters per hectare per year in the growing stock.

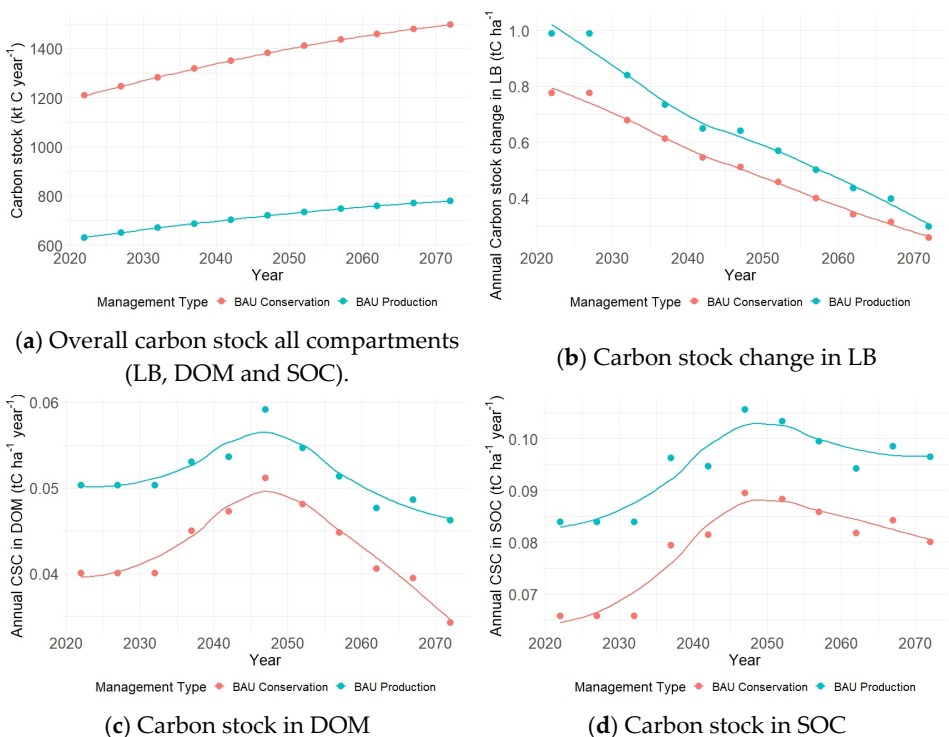

(**a**) Overall carbon stock all compartments (LB, DOM and SOC).

(**b**) Carbon stock change in LB

(**c**) Carbon stock in DOM

(**d**) Carbon stock in SOC

**Figure 7.** Carbon stock distribution and change in PCNP's forest ecosystems: 2022–2072 ((**a**)—Overall carbon stock all compartments (LB, DOM and SOC), (**b**)—Carbon stock change in LB, (**c**)—Carbon stock in DOM, (**d**)—Carbon stock in SOC). The line in the figure represents a summary of trends within the data, employing locally estimated scatterplot smoothing (LOESS), a non-parametric method for smoothing scatterplots.

In both management scenarios (conservation and production), the carbon stock is set to increase over a span of 50 years. However, while they remain carbon sinks, the annual carbon stock change—which quantifies the difference between two successive yearly measurements—is decreasing in both scenarios. This trend is attributed to a reduction in the growth rate and, on the other hand, an increase in mortality and harvest rates as the forests age. In particular, the projections indicate a reduction, transitioning from an initial rate of 11,000 metric tonnes per year (equivalent to 0.8 metric tonnes of carbon per hectare per year) to a final rate of 3000 metric tonnes per year (corresponding to 0.3 metric tonnes of carbon per hectare per year) (Figure 7b). However, when examining the carbon stock change in dissolved organic matter (DOM) and soil organic carbon (SOC), the data revealed an initial upward trajectory for both scenarios (Figure 7c,d). Nevertheless, starting from 2046 onwards, these values began to exhibit a decline, with a more pronounced reduction observed in the case of carbon stock in DOM (Figure 7c).

**Annual trends in carbon stocks and carbon stock change comparison between future harvest scenarios.**

The comparative analysis of the two scenarios reveals significant insights into the dynamics of the annual carbon stock change in all biomass compartments (Figure 8a). The BAU scenario shows a noticeable reduction in the annual carbon stock change, declining from 0.8 t C ha$^{-1}$ to 0.3 t C ha$^{-1}$. Similarly, in the MIN scenario, the annual carbon stock change experiences a decrease from 1.45 t C ha$^{-1}$ to 1.0 t C ha$^{-1}$. The MAX scenario presents a contrasting pattern in terms of harvest activities. This scenario exhibits a consistent increase in the annual carbon stock change due to higher harvest rates, particularly during

thinning operations for the next 30 years, with 25%, and a decrease after reaching a slightly lower CSC value as from the simulation's start (Figure 8b).

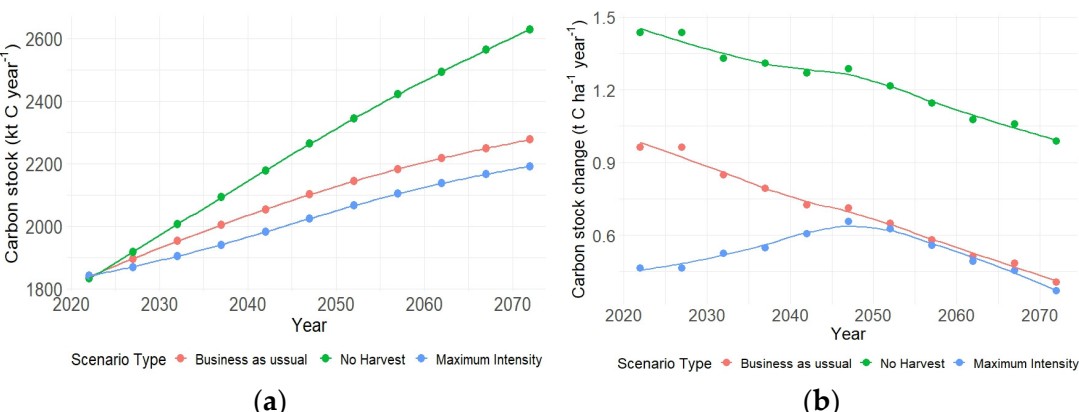

**Figure 8.** (**a**) Temporal trends in PCNP forests across three scenarios (BAU, MIN, and MAX) from 2022 to 2072. (**b**) Annual carbon stock change in all C pools (LB, DOM and SOC) in PCNP forests across three scenarios (BAU, MIN, and MAX) from 2022 to 2072.

Regarding the carbon stock in the dead matter within the research area, the developed model exhibits a consistent upward trajectory over the forthcoming two decades (Figure 8b). This trajectory is anticipated to lead to a state of equilibrium, observed in both the MAX and MIN scenarios, in terms of the annual quantity of carbon sequestered. Concurrently, in the Business as Usual (BAU) scenario, the opposite trend is identified, where the annual quantity of carbon sequestered in the dead matter, encompassing dead wood (DW), dissolved organic matter (DOM), and soil organic carbon (SOC), is projected to decline during this same period. This shift in carbon dynamics highlights a divergence in the BAU scenario compared to the MAX and MIN scenarios, underscoring the potential ecological implications of various management approaches on the carbon balance within the ecosystem (Supplementary S2).

Regarding the potential revenue streams associated with carbon storage or carbon sink (credits in carbon), in the analysis focusing on the forests within PCNP within the MAX scenario, it is anticipated that the annual revenue generated from $CO_2$ sequestration will lead to the highest financial returns, amounting to EUR 4.1 million per year. In the most plausible scenario, namely the BAU scenario, average annual revenue of EUR 2.3 million is projected to be generated each year. These revenue projections underscore the potential economic benefits of carbon storage initiatives within forest ecosystems and the varying financial outcomes achievable through different management scenarios. If the MIN scenario is implemented, the model employed in this study predicts an annual reduction of 23,000 cubic meters ($m^3$) in wood harvested. Over the next five decades, this cumulative loss in annual harvest income is projected to amount to 1.2 million $m^3$ of wood. Considering an average market price of EUR 60 per $m^3$ of firewood, it can be extrapolated that an annual income of approximately EUR 1.4 million will be generated. Estimating the cumulative value derived from the sale of firewood within the MIN scenario reveals an approximate total of EUR 69 million over the entire 50-year scenario period. This calculation underscores the financial implications of wood production and highlights the potential economic outcomes of the MIN scenario.

Furthermore, it is worth noting that the MIN scenario is expected to augment the annual revenue from wood production substantially. Specifically, this scenario is projected to boost the yearly income by 40%, resulting in a total of EUR 2 million per year (Table 2). This increase reflects the economic advantages associated with the MIN scenario's approach to wood production and underscores its potential impact on the financial sustainability of forest management strategies in the PCNP.

**Table 2.** Carbon stock change in AGB and dead matter, annual capture capacity, and annual revenue (2022–2072).

| Scenario Type | Mean Net Carbon Stock Change per Area in AGB (tC/ha/year) | Mean Annual $CO_2$ Captured (t $CO_2$) | Mean Annual Wood Production Harvest (m$^3$) | Mean Annual Revenue in Carbon Credits (EUR) | Mean Annual Revenue in Wood Harvest (EUR) | Total Revenue (EUR) |
|---|---|---|---|---|---|---|
| **BAU** | 0.69 ± 0.18 | −29,025 ± 7507 | 23,339 ± 3876 | 2,322,019 ± 600,539 | 1,400,319 ± 232,584 | 3,722,338 |
| **MIN** | 1.23 ± 0.14 | −51,508 ± 5981 | 0.00 | 4,120,652 ± 478,464 | 0.00 | 4,120,652 |
| **MAX** | 0.52 ± 0.08 | −21,887 ± 3395 | 32,844 ± 1432 | 1,750,975 ± 271,626 | 1,970,659 ± 85,940 | 3,721,634 |

## 4. Discussion

Forest management in Romania is built on multiple sustainability principles, with the most common being the principle of sustained forest yields, established in 1954 [60]. Forests in Romania are generally classified into two main categories: those with special protection functions and those with both production and protection functions [61–63]. According to Romanian legislation [64], all forest areas must be managed in accordance with a forest management plan (FMP), and all forest-related activities are subject to uniform management practices, independent of ownership or size [65,66]. The national legislation permits a variety of forest management systems, including high-forest systems and coppice, among others. However, the most prevalent management system in Romania is the high-forest system, which typically employs group shelterwood and uniform shelterwood cuttings, with an emphasis on natural regeneration [63,67,68]. Production cycles, or rotation ages, are generally lengthy (over 100 years), varying based on the tree species and wood assortments. The longer rotation ages are mainly characteristic of forests that fulfil both production and protection functions, thereby aligning with the principles of sustainable forest management [21].

To align with EU forest legislation, such as the Green Deal, the EU Biodiversity Strategy 2030, and the new EU Forest Strategy for 2030, Romanian forestry legislation has undergone several revisions and it will have further modifications in the near future. These changes aim to satisfy the diverse needs of users under the concept of sustainability [65]. Both EU and national attention is increasingly focused on the conservation of forest areas, especially in primary and old-growth forests. There have been debates around the conflicting definitions linked to the specific context of each country [69], highlighting the need for these forests to be placed under strict protection. To effectively navigate the trade-off between conservation and production in forests, there is a growing focus on utilizing modeling approaches that integrate mitigation models and wood production analysis. This approach helps in making informed decisions that balance conservation objectives with the demand for forest products.

Our study compares forests with different functions, such as the sustainable conservation zone (only special conservation cuttings are allowed) and the sustainable development zone (silvicultural systems that promote the natural regeneration of the stands and respect the principles of the sustainable use of natural resources are permitted). Stands' gross growth and vitality serve as indicators of a good, balanced ecosystem in the Piatra Craiului Park, offering benefits beyond carbon sequestration. These benefits encompass support for biodiversity, improvements in soil health, and an overall enhancement in ecosystem services.

The age class distribution across areas reveals an imbalance, with a skew towards older stand ages in conservation management areas, where most of the area comprises stands over 100 years old. In contrast, production management areas predominantly feature

stands in their most productive phase, around 60 years old. This highlights the significance of preserving and sustainably managing forests during their prime productive stages, achieving the maximum ecological and environmental benefits. Although the conservation management type covers a larger area (Figure 4), the growth per ha is significantly higher for forests under production management. The irregular age distribution in conservation stands is largely due to the lack of silvicultural interventions, such as logging, with changes occurring mainly through natural mortality. On the other hand, the consistent age structure in production stands results from ongoing management practices and harvesting activities. These practices are designed to maintain a balanced age distribution within the forest stands, promoting a sustainable and productive forest ecosystem. Under both management approaches, the growing stock is projected to increase with the same margin by over 40% in the next 50 years according to the most probable scenario, Business as Usual (BAU). However, slight variations, such as an increased trend in growing stock for conservation stands, are noticeable. These subtle differences are further emphasized by the reduced yield in conservation areas (as shown in Figure 2B) and elevated mortality rates, which counterbalance the management practices in production regions.

Based on historical harvest intensities and age class dynamics, the yield is generally anticipated to decrease on average, with the exception of beech species in conservation plots, where a slight increase is projected. However, for mixed stands in the same plots, the yield is expected to rise in the next 20 years. Overall, in the Business as Usual (BAU) scenario, there is a consensus across all forest stands that the yield is likely to decrease. This decline is anticipated alongside an increase in mortality and harvest, most likely due to the aging of the forests.

In terms of carbon stock, an overall increase is expected due to the growth in tree biomass. However, for both scenarios, the carbon stock change—measured as the difference between gains and losses—will likely decrease the annual net change in living biomass. Conversely, an increasing trend is expected in dead organic matter (DOM) and soil organic carbon (SOC) compartments for the next 20 to 30 years, followed by a decrease until the end of the modeling exercise.

The findings of this study, based on the three distinct forest management scenarios—Business as Usual (BAU), Maximum Intensity (MAX), and No Harvest (MIN)—shed light on the complex interplay between forest management, carbon sequestration, and economic outcomes in the context of the PCNP. Under the BAU and MIN scenarios, a reduction in carbon stock change (CSC) is observed. These trends are primarily attributed to the aging of the forest stands, leading to naturally reduced carbon sequestration rates. In the MIN scenario, which adopts a conservative approach to timber harvesting with low rates, there is a notable doubling of annual carbon storage from tree biomass. This indicates a significant annual increase in the forest's carbon sequestration capacity, highlighting the ecological benefits of reduced timber extraction and the positive impact on carbon accumulation in the forest's living biomass. In the MAX scenario, characterized by heightened harvesting activity, forest growth is stimulated over the subsequent two decades. However, this initial growth phase is followed by a decline, bringing the annual carbon stock change back to levels similar to the BAU scenario (Figure 8b). Overall, the simulation results from the entire modeling exercise reveal that the forest stands within the PCNP are currently in their most productive phase. This critical phase is marked by optimized growth rates and carbon sequestration capacity and overall ecosystem vitality.

To achieve net-zero emissions, the EU recommends nature-based solutions like forest carbon sequestration [70]. This creates a balance between maximizing carbon stocks and wood production, which requires significant financing [71]. While timber production remains economically vital [72], certain management practices can either increase wood production at the expense of carbon stocks, like clear-cutting [10], or enhance carbon stocks but reduce wood yields, such as afforestation [73,74].

The study's findings have broader implications for forest management policies, not only within the PCNP but also in similar forested regions. They underscore the need for a

balanced approach that considers both ecological and economic factors. The implementation of the BAU scenario could serve as a model for sustainable forest management, demonstrating how economic viability can be aligned with climate change mitigation goals.

While our study provides valuable insights, it is not without limitations. The EFISCEN model, although robust, is based on certain assumptions and parameters that might not capture the full complexity of forest dynamics. Mainly, the model is designed for large forest areas; thus, several factors may be considered in smaller regions, such as the data resolution and local model calibration. The model's predictions might be more sensitive to data inaccuracies and assumptions at a smaller scale. To fully understand the complexity of forest dynamics, a future study must consider potential climate changes, such as temperature variations, drought, and excessive pollution. Additionally, the study should account for forest fires and the resulting impacts of these climate changes, including threats from pests and diseases. While these factors can be seen as limitations in our current research, it is important to mention that our model, used in this study, did not incorporate these aspects. In this initial stage, our focus was on showcasing the evolution of the studied forests under existing conditions within the scope of this case study. Further research should expand to a broader area, incorporating these considerations and parameters to provide a more comprehensive analysis. Moreover, future research should consider incorporating more variables, such as biodiversity indices and soil carbon data, to enhance the model's predictive power. Another potential factor to consider in constructing a new scenario model is illegal logging. However, we currently believe that this aspect may not significantly impact the progression of the forests. This is because concerns about such issues have recently heightened in Romania, leading to the establishment of a comprehensive monitoring system for wood mass at the national level. This system encourages participation from all stakeholders, including authorities, forest owners and managers, citizens, and NGOs. By gathering information collaboratively, the system aims to minimize the adverse effects of illegal logging (Iordache and Derczeni 2018; Vasile and Iordachescu 2022). Despite this, utilizing historical information from this system could enable the development of a forecast model based on actual data.

Moreover, the study's focus on a 50-year timeline provides a medium-term perspective. Long-term studies extending beyond this period could offer additional insights into the sustainability and effectiveness of the different management scenarios.

Adopting a no-harvest management system in the MIN scenario could reduce carbon emissions from firewood burning in the region [75]. This might increase the reliance on carbon-emitting fossil fuels like coal, oil, or natural gas to meet energy demands [76]. Moreover, a decreased fuelwood demand in one area could unintentionally boost the demand in others, potentially leading to deforestation and increased emissions due to overexploitation, especially in regions with lax environmental regulations or enforcement. Therefore, while reducing firewood usage could lower emissions locally, the broader indirect effects and unintended consequences must be considered. A comprehensive approach, considering the entire supply chain and global market dynamics, is essential for effective carbon emission reduction and climate change mitigation.

## 5. Conclusions

In conclusion, our study highlights the significance of nuanced forest management in achieving a win–win relationship between carbon sequestration and economic viability. The BAU scenario, with its balanced approach, offers a promising pathway for sustainable forest management in the PCNP and potentially other similar regions. It is imperative that policy decisions and management practices are informed by such comprehensive analyses to ensure that the goals of climate change mitigation, the protection of forest areas, and economic sustainability are met.

Compensation for harvested forests is also essential, as it can incentivize landowners to manage their forests in a sustainable manner, for the use of sustainable harvesting techniques, selective logging, or reforestation, which preserve the ecological integrity of the

forest and promote its long-term health status. This can help to ensure that forests continue to provide important ecological and economic benefits over time. Clear and enforceable legislation for compensation for both non-harvested and harvested forests can help to mitigate conflicts between stakeholders (forest owners/associations, policymakers, citizens, ENGOs).

**Supplementary Materials:** The following supporting information can be downloaded at: https://www.mdpi.com/article/10.3390/land13010017/s1, Supplementary S1: Average wood prices provided from sources: EU-EUTS and RNSI; Supplementary S2: Model initialization data and model results.

**Author Contributions:** Conceptualization, S.C. and R.G.R.; methodology, S.C., R.G.R., F.C., L.M. and S.P.; software, R.G.R., F.C. and S.P.; validation, R.G.R., M.H., D.P., S.L. and O.B.; formal analysis, S.C., F.C., M.H., D.P., L.M. and S.P.; investigation, S.C., F.C., D.P., S.L. and O.B.; resources, S.C., R.G.R., S.L. and O.B.; data curation, R.G.R., F.C. and S.L.; writing—original draft preparation, S.C., R.G.R., F.C., M.H. and L.M.; writing—review and editing, S.C., R.G.R., F.C., M.H., D.P., L.M., S.L. and O.B.; visualization, D.P., S.L. and O.B.; supervision, S.C., D.P. and O.B.; project administration, S.C., R.G.R., D.P. and O.B.; funding acquisition, R.G.R., D.P. and O.B. All authors have read and agreed to the published version of the manuscript.

**Funding:** This research was made possible through the funding provided by the Romanian Ministry of Research, Innovation and Digitalization, FORCLIMSOC Nucleu Programme (Contract 12N/2023), project PN 23090202; partially through the project "Creșterea capacității și performanței instituționale a INCDS 'Marin Drăcea' în activitatea de CDI—CresPerfInst" (Contract 34PFE/30.12.2021); partially by the project CRESFORLIFE (SMIS 105506), subsidiary contract no. 19/2020, co-financed by the European Regional Development Fund through the 2014–2020 Competitiveness Operational Program; and partially by contract 47/N/2019 between the Ministry of Waters and Forests and the National Institute for Research and Development in Forestry "Marin Drăcea" in supporting activities for estimating and reporting national greenhouse gas emissions and retention in forest land use as part of the Land Use and Land Use Change and Forestry sector.

**Data Availability Statement:** The data presented in this study are available in [Supplementary S2].

**Acknowledgments:** The authors express their gratitude to their colleagues in the Department of Forest Management from INCDS and the foresters from the Piatra Craiului National Park, as well as to all individuals involved in the research.

**Conflicts of Interest:** The authors declare no conflict of interest.

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
