# Peer review of "Cost Valuation and Climate Mitigation Impacts of Forest Management: A Case Study from Piatra Craiului National Park, Romania"

_land, doi:10.3390/land13010017_

Round 1

Reviewer 1 Report

Comments and Suggestions for Authors

Thank you, editor and author(s), for providing an opportunity to read this manuscript entitled “Cost valuation and climate mitigation impacts of forest management: A case study from Piatra Craiului National park, Romania”. The manuscript is well-written. However, there are a few things that need to be addressed.

Comments:

Lines 154-157 reported about the data collection for the study. However, the manuscript fails to provide sufficient information about how those data were collected. Also, I suggest that the author(s) provide the descriptive statistics of each of the data (or variables) in the results section.

Line 160: What were the tools and techniques used for field observations and measurements?

Line 275: There are many sources to obtain the price for carbon credit. Why did the authors use the European Energy Exchange (EEX) in the study? Please provide a couple of reasons.

Line 336-338: Grammatical error. The sentence needs to be rearranged.

Line 347-348: The author(s) uses metric tons and tonne interchangeably. Be advised to use consistency throughout the manuscript.

Figure 7b and c: Grammatical error in caption. Need an upper case for the initial words.

Line 458-460: “Under both management approaches, the growing stock……..” It seems production and conservation approaches did not have any effect on growing stock in the future. Is it true? Theoretically, the conservation forest could have a lower growing stock than the production forest in the long run. It is because the increasing growing stock in conservation forests is risky for wildfire, diseases, or insects. So, please provide the appropriate reason with a citation to support your statement.

Author Response

Dear Reviewer,

We sincerely appreciate the time and effort you dedicated to reviewing our manuscript. Your feedback has been well received and thoroughly considered. In response, we have meticulously revised the document, incorporating your suggestions to enhance its scientific value and rigor.

We have attached a document with all the updates to the manuscript according to your comments.

Reviewer 2 Report

Comments and Suggestions for Authors

The manuscript is well-written and relevant to the growing need for more studies on carbon valuation worldwide. The Introduction captured what needs to be done to address the challenges in the economic valuation of carbon. It presents clearly the objectives and research questions. The Methods described the model extensively so readers can be guided if they wish to conduct similar studies. The analysis is robust. The Results are well-presented, with meaningful graphics and tables that are easy to understand. The Discussion section was well-written, backed up by recent scientific references. It also presents implications with pros and cons and ends with an insightful take-home message for the readers.   

I have only a few minor comments:

1.       Line 26-27: the 3 in “m3” must be a superscript

2.      Line 58: “4,6%” into “4.6%”

3.      In the Introduction, please present the novelty of this study and some hypotheses.

4.      The Piantra Craiului National Park has been mentioned many times in the paper. It might be better to spell this out from the first mention and then abbreviate the succeeding ones.

5.      Figure 1: Map legends are too small to read.

6.      Line 126: EFISCEN model – describe the referecnes, accessibility, and other information so that readers may know where to access it if they want to use this model.

7.      Section 2.3: Any information how recent are the data used from the archives? Maybe a table that would show the averages of these variables in the model? What are your data inputs? Thresholds or anything that will serve as a reader reference aside from Table 1?

8.      Lines 263-265: Also briefly describe the MAX and MIN scenarios.

9.      Figure 2B and in other figures: In y-axis unit “m3/ha” “3” must be a superscript or m3 ha−1

10.   Figure 3B:  In y-axis unit “m3” “3” must be a superscript

11.   Figure 4 caption: “panne;” spell check

12.   Lines 344-345: “This trend is attributed to a reduction in growth”

     Is this a growth reduction per se or a reduction in growth rate?

13.   Lines 347-349: This sentence is vague. Consider revising it to make it more understandable. “Specifically”… what is being specified here?

Author Response

(The authors gave the same response as above.)

Reviewer 3 Report

Comments and Suggestions for Authors

Thank you for preparation of interesting manuscript regarding carbon sink in protected forest stand.

In Introduction chapter, line 59 please insert how many workers are a part of forestry and wood sector together in Romania (in thousands).

In subchapter line 119 please explain which kind of forest management plans exist in the National park. Who is responsable for forest management. Which kind of intensity is allowed to cut in protected area. Which type of forest management is prescribed (even age forest or univen aged forests). Are there some of Natura2000 sites or ecological network. How big is average growing stock per hectare, and percent of annual cut. Regarding the statement in line 123 please explain in which kind of protected area is active forest management to achieve suatsainability goals.

In chapter 2.2. line 126 please explain whay EFISCEN model is the most suitable for this case study area. Is this model more suitable for conifers, broalives or mixed forests. Or there is no obstacles regarding the type of forest. Model predictions could have significant change based on various input parameters.

In chapter 2.3. line 146 regarding the history please insert when was the first established management plan in selected National park.

Among the other characteristics in line 151, did you register stand quality level.

In line 172 it would be better to write 11 400 ha (not kha).

In line 248 please insert the main national legislation and authority is it Forest Law or Law on Nature Protection and responsible Ministry.

In line 257 are the mentioned years for the final cut prescribed rotation age  for stand or for each species.

In chapter Discussion line 506 please xplain which kind of EFISCEN model  limitations could have the most significant influence in your case. PLanning the forest management in future and carbon sink should consider climate change risk, what could have significant influence in growing stock and increment.

Author Response

(The authors gave the same response as above.)

Reviewer 4 Report

Comments and Suggestions for Authors

This is a mostly sound and interesting study. There are however a few issues that need to be addressed to bring it up to the required standard.

First and most importantly, the study is fairly comprehensive in the variables considered but fails to consider two. One is the possible impacts of climate change itself on forest dynamics. Rising temperatures, drought and/or increased wildfires in the 50 years ahead could directly and negatively affect forest growth and stock, and/or alter the mix of tree species according to their respective tolerances to different climatic/soil conditions. Other, secondary effects of climate change (such as increased pest/disease threats) could also have an impact. By the authors' own admission, the model "is based on certain assumptions and parameters that might not capture the full complexity of forest dynamics" (lines 507-508). Furthermore, these parameters are assumed to be stable throughout the 50-year period modelled. It seems however to be a serious omission (in a study premised on mitigating climate change) not to at least mention how climate change impacts may disrupt the parameters set and briefly discuss this, even if such impacts are not directly factored into the model on this occasion. (Along with other variables, they can reasonably be identified as a subject for further study.)

The other missing factor is the extent to which forests are managed according to state regulations. This is only hinted at in places (for example in the Conclusions) but again deserves brief discussion as it relates directly to the credibility of the different scenarios described. Sadly, as the authors must be aware, illegal logging is a problem in Romania's Southern Carpathians (as recently reported in the UK's Guardian newspaper: https://www.theguardian.com/travel/2023/oct/22/a-yellowstone-for-europe-romanias-ambition-for-a-vast-new-wilderness-reserve).

Second, even within the parameters used, there seems to be a disparity in the local average temperature used in the model. It is stated (lines 116-117) that "the average annual temperature has values between 0-5 °C, values assessed for 1961-2000" but later that the model works on "the local average annual temperature at 7.3°C" (lines 239-240). Even accounting for a warmer 21st century to date, this does not make sense unless I am missing something.

Comments on the Quality of English Language

Finally, the prose needs tidying up throughout the text, please. It is rather wordy overall (for example, it is unnecessary to use phrasing like "It is well known that..." (lines 46-47); the authors need simply to evidence such statements with citations and/or their own research). Elsewhere the prose reads incorrectly (for example it should be "gases", not "gasses", where these are mentioned) or awkwardly to a native English speaker, which a careful edit should remedy. The first two lines of the Results section (lines 285-286) are also not written as sentences but as notes, which looks like an oversight before submission.

I very much hope that these comments are helpful to the authors in improving their otherwise commendable paper.

Author Response

(The authors gave the same response as above.)
